Cryophysiology of coral microfragments: effects of chilling and cryoprotectant toxicity

Lager Claire V. A. 1 2 lagerc@hawaii.edu
Perry Riley 1 2
Daly Jonathan 1 3 4
Page Christopher 1 2
Mizobe Mindy 2
http://orcid.org/0000-0002-9221-537X Bouwmeester Jessica 1 2
Consiglio Anthony N. 5
Carter Jake 5
Powell-Palm Matthew J. 6 7
Hagedorn Mary 1 2
1 Center for Species Survival, Smithsonian’s National Zoo and Conservation Biology Institute , Front Royal, VA , United States
2 Hawaiʻi Institute of Marine Biology , Kāne’ohe, HI , United States
3 Taronga Conservation Society Australia , Mosman, NSW , Australia
4 School of Biological, Earth and Environmental Science, University of New South Wales , Sydney, NSW , Australia
5 Department of Mechanical Engineering, University of California, Berkeley , Berkeley, CA , United States
6 J. Mike Walker ʻ66 Department of Mechanical Engineering, Texas A&M University , College Station, TX , United States
7 Department of Materials Science and Engineering, Texas A&M University , College Station, Texas , United States
Eamens Andrew
Electronic publication date: 2024 Nov 11
Publication date: 2024
Volume: 12
Electronic Location ID: e18447
Received 2024 Mar 22; Accepted 2024 Oct 14
Copyright: © 2024 Lager et al.
Copyright year: 2024
Copyright holder: Lager et al.
License: This is an open access article distributed under the terms of the Creative Commons Attribution License, which permits unrestricted use, distribution, reproduction and adaptation in any medium and for any purpose provided that it is properly attributed. For attribution, the original author(s), title, publication source (PeerJ) and either DOI or URL of the article must be cited.
License URL: https://creativecommons.org/licenses/by/4.0/

Keywords: Porites compressa, Confocal imaging, Green fluorescent protein, Microfragment

Funding: Revive & Restore Catalyst Science Fund 2023-049 Smithsonian Institution Hawaii Institute of Marine Biology The Smithsonian’s Women’s Committee Paul M. Angell Family Foundation OceanKind Scintilla Foundation Zegar Family Foundation William H. Donner Family Foundation Anela Kolohe Foundation Cedar Hill Foundation Funding was provided by the Revive & Restore Catalyst Science Fund to Mary Hagedorn and Matthew Powell-Palm (2023-049). Additional funding was provided to Mary Hagedorn by the Smithsonian Institution, the Hawaii Institute of Marine Biology, The Smithsonian’s Women’s Committee, the Paul M. Angell Family Foundation, OceanKind, the Scintilla Foundation, the Zegar Family Foundation, the William H. Donner Family Foundation, Anela Kolohe Foundation and the Cedar Hill Foundation. The funders had no role in study design, data collection and analysis, decision to publish, or preparation of the manuscript.

==============================
Coral reefs are being degraded at alarming rates and decisive intervention actions are urgently needed. One such intervention is coral cryopreservation. Although the cryopreservation of coral sperm and larvae has been achieved, preservation of coral fragments including both its tissue and skeleton, has not. The overarching aim of this study was to understand and assess the physiological stressors that might underlie coral fragment cryopreservation, understand the long-term consequences of these exposures to continued growth, and develop a health metrics scale for future research. Therefore, we assessed small fragments (~1 cm2) from the Hawaiian coral, Porites compressa, examining: (1) chill sensitivity; (2) chemical sensitivity to complex cryoprotectants; (3) methods to safely remove algal symbionts of coral for cryopreservation; (4) continued growth over time of coral fragments exposed to chilling and cryoprotectants; and (5) assessment of health and viability of coral fragments post the applied treatments. Corals were able to withstand chilling to 0 °C for 1 min and after 2 weeks were not significantly different from the live controls, whereas, corals exposed to complex cryoprotectants needed 3 weeks of recovery. Most importantly, it appears that once the coral fragments had surpassed this initial recovery, there was no difference in subsequent growth. Technological advances in cryo-technology promise to support successful coral fragment cryopreservation soon, and its success could help secure much of the genetic and biodiversity of reefs in the next decade.

Introduction

The coupling of climate change and anthropogenic stressors has caused a widespread and well-recognized reef crisis (Bellwood et al., 2004; Madin & Madin, 2015; Eakin, Sweatman & Brainard, 2019; França et al., 2020; Frölicher, Fischer & Gruber, 2018). New modeling data suggest that the threat to tropical coral reefs may be challenging even with the most optimistic assumptions of coral reef refugia, adaptation, and potential for restoration with near total reef loss expected by the middle of this century (Dixon et al., 2022; Kalmus et al., 2022). As part of these stressors, ocean warming is increasing the frequency of bleaching events around the world (Hughes et al., 2018), which has been shown to negatively impact coral reproduction (Hagedorn et al., 2016; Ward, Harrison & Hoegh-Guldberg, 2002; Henley et al., 2021). Without robust reproduction on reefs, the potential for adaptations to warmer waters is reduced (van Oppen et al., 2015). Innovative and practical conservation solutions are needed to help preserve coral genetic diversity and biodiversity. Decisive conservation actions are urgently needed to save our reefs.

Cryopreservation is a state-of-the-art tool that has been used successfully for decades to preserve genetic and biodiversity in many wildlife species (Wildt, 1992; Prieto et al., 2014; Comizzoli, 2017). The process works because, through a series of steps, water within the cell is extracted and replaced with a cryoprotectant or antifreeze. The partially dehydrated cell can then withstand the extraordinary stress of low temperature exposure, essentially entering a state of suspended animation (Mazur, 1984). Cryopreservation can maintain the sample cold-but-alive for decades, thus offering much needed time to help resolve in situ conservation challenges. Cryopreservation is a maturing conservation tool that already has impressive milestones for coral. To date, the global community has cryopreserved coral sperm from over 50 species (Smithsonian’s National Zoo & Conservation Biology Institute, 2021). These cryopreserved assets have been used to create embryos from these frozen sperm samples for restoration and assisted gene flow (Daly et al., 2022; Hagedorn et al., 2012, 2017, 2021).

However, sexual reproduction occurs for most coral species only over a few days each year (Babcock et al., 1986; Bouwmeester et al., 2021), and much of this reproductive material is faltering in some areas of the world due to stressors (Randall & Szmant, 2009; Levitan et al., 2014; Hagedorn et al., 2016). Because coral sexual reproduction will likely continue to be negatively impacted and be of uncertain quality, an important milestone in coral cryopreservation is to preserve small pieces of asexually reproduced adult coral (1 cm2), also commonly called coral ‘microfragments’ (Koch et al., 2021; Page, Muller & Vaughan, 2018). This strategy of cryopreservation would be independent of sexual reproduction both before freezing (successful spawning, sperm motility, and larval development) and after post-thaw (fertilization, settlement) and could be accomplished throughout the year.

However, before robust cryopreservation strategies for coral microfragments can be developed, basic cell sensitivities to chilling and cryoprotectant solutions must be tested, and the response of these microfragments to these stressors must be monitored over time. To our knowledge, the only other whole, adult organism to be cryopreserved and successfully revived is the nematode, Caenorhabditis elegans (Hayashi et al., 2013). Not only is it important to produce viable cryopreserved coral, but it is equally important to create a clear husbandry pathway to return these microfragments to a land-based nursery setting post-thaw. This study examined: (1) the sensitivity of the coral microfragments with their algal symbionts to chilling temperatures; (2) the response of the coral to complex cryoprotectant cocktails in terms of toxicity, (3) how long it took them to start regrowing after this exposure; (4) methods to safely remove the algal symbionts from the coral fragment before cryopreservation resulting in bleached microfragments, given the two symbiotic partners have very different membrane permeabilities to water and cryoprotectant (Hagedorn et al., 2009, 2006); and (5) a method to develop a visual scale of health metrics to assess damaged corals. Methods for quantifying physiological responses to these stressors include confocal imaging, pulse amplitude modulated fluorometry, imaging pulse amplitude modulated fluorometry, light microscopy, standardized health metrics and bleaching color cards. A deep understanding of these types of detailed physiological stressors and metrics will be critical to help overcome the inevitable stress of cryopreserved coral microfragments. Portions of this text were previously published as part of a preprint (https://doi.org/10.1101/2023.01.03.522625).

Materials and Methods

Coral collection and microfragmentation

Porites compressa is an endemic, reef-building coral that is prolific around the State of Hawaiʻi. Colonies were collected from various reefs throughout Kāne‘ohe Bay, O‘ahu, Hawai‘i (21.4, −157.8) in accordance with our collecting permit from the Department of Land and Natural Resources from the State of Hawai‘i (Special Activity Permit #2022-22 from the Hawai‘i Institute of Marine Biology). A hammer and chisel were used to collect 10−15 cm portions from individual colonies. Colonies were collected at least 5 m apart on each patch reef and from several patch reefs throughout Kāne‘ohe Bay to avoid collecting clones of the same genotype. Once collected, colonies were kept in outdoor aquaria with a filtered, flow-through seawater system at the Hawai‘i Institute of Marine Biology on Moku o Lo‘e. Colonies were microfragmented as soon as one day after collection and up to several weeks after.

Prospective colonies were progressively fragmented with a bandsaw (Gryphon XL AquaSaw Diamond Band Frag Saw, http://www.gryphoncorp.com/) to yield uniformly sized microfragments (1 cm2) and glued (BRS Extra Thick Gel Super Glue–Bulk Reef Supply) onto a plastic sheet supported by a plexiglass plate (Page, Muller & Vaughan, 2018). Microfragments were then allowed to heal for 2 weeks prior to experimentation.

Vitrification solution preparation

The solutions in this study were derived from previous methods and solutions used to cryopreserve coral larvae (Daly et al., 2018) which utilized a cryopreservation process known as vitrification (Sakai & Engelmann, 2007)–this technique avoids lethal ice formation by allowing the liquid within the system to enter a vitreous or glassy state by using high concentrations of solutes and ultra-rapid cooling and warming. Given the size and complexity of coral microfragments, future preservation of coral microfragments will likely require a cryopreservation process called isochoric vitrification, in which isochoric (constant-volume) conditions are used to decrease the likelihood of ice formation in the system, thereby enabling vitrification at lower, less toxic concentrations (Rubinsky, Perez & Carlson, 2005). Anticipating using this cryopreservation modality, the cryoprotectants in the vitrification solutions used in this study were reduced 20–24% (by mass) and the trehalose was reduced by 23% (by mass). Two different strengths of the same vitrification solution (VS.) were prepared for testing: (1) VS80 (0.8 M dimethyl sulfoxide, propylene glycol and glycerol, 0.7 M trehalose in 0.3 M PBS (phosphate buffered Saline); and (2) VS76 (0.75 M dimethyl sulfoxide, propylene glycol and glycerol, 0.7 M trehalose in 0.3 M PBS, see Table S1 for details).

Chilling and toxicity were not tested together, because during cryopreservation as the tissue starts to chill, the permeability of the cell membranes reduces the flow of cryoprotectant into the cells, thus limiting the impact of toxicity. Vitrification would be necessary for large complex tissues like coral microfragments, and generally, vitrification procedures involve dehydrating and equilibrating tissues in their vitrification solution at room temperature and then rapidly cooling the coral to cryopreservation temperatures.

Toxicity of vitrification solution to coral microfragments

To determine how long microfragments could be exposed to the vitrification solution, preliminary toxicity experiments were conducted over several time points (1 to 5 min). Because the means were not different across exposure times (p > 0.05; Fig. S2), we continued our experiments with a slightly longer exposure of 6 min and extended the time we monitored recovery from 2 weeks to 3 weeks.

Coral microfragments were placed in vitrification solution of either VS76 (n = 11) or VS80 (n = 11) for 6 min in one step and then placed through a rehydration series (Table S2). The physiological effect of the solutions on the coral were not significantly different (Mann-Whitney test, p > 0.05). Thus, the results from these two vitrification solutions were eventually pooled as one treatment, called VS. Coral microfragments were placed into individual 6-well plates with approximately 10 mL of 0.22 microliter (μL) filtered seawater (FSW) for recovery. Six-well plates were then placed in an incubator (26 °C), covered in aluminum foil to avoid any photosynthetically active radiation (0 PAR) for the first 24 h post-cryoprotectant exposure, and kept in very low light (35–50 PAR), thereafter. Microfragments were assessed at five time points: 24 and 72 h, 1-, 2-, and 3-weeks post-cryoprotectant exposure. Each assessment included: Junior PAM fluorescence reading, laser-scanning confocal microscopy imaging, light microscopy imaging and health metric scoring (Table 1). A health score of 0–5 was assigned to each coral microfragment at each of the health assessment time points (24 and 72 h, 1, 2, and 3 weeks). The health score was developed based on the following criteria: tissue loss and algal symbiont loss, tissue color, damaged or intact polyps, and intact coenosarc (Table 1). Microfragments were kept in 6-well plates with FSW (changed daily) in an incubator (26 °C) through the 2 week assessment, after which they were placed in 5 L aquaria with flow-through seawater until the 3 week assessment.

Table 1 Health metric scoring criteria for unbleached microfragments.

Coral fragments that had not been bleached were observed visually under a light microscope and given a score 0–5 using the criteria listed here.

Score	Description of metric for unbleached microfragments	
0	Dead, coral tissue and algal symbionts released from skeleton, yellow/green color	
1	Damaged polyp, green color, <10% intact coenosarc	
2	Damaged polyp, pale, 25% intact coenosarc	
3	Intact polyp, paling color, 50% intact coenosarc	
4	Intact polyp, normal tissue color, 75% intact coenosarc	
5	Intact polyp, normal tissue color, 100% intact coenosarc	

Pulse amplitude modulation fluorometry

For these studies we used two types of Pulse Amplitude Modulated (PAM) fluorometer. The first was a Junior-PAM (Walz, Effeltrich, Germany) with a single fiber optic cable of approximately 1 mm in diameter for the toxicity and chilling experiments. The second was an Imaging-PAM (IMAG-MAX/L; Walz, Effeltrich, Germany) for the bleaching experiments. Both PAM units measure functionality of Photosystem II in photosynthetic organisms. Specifically, photosynthetic yield was used to determine the approximate health and functionality of the algal symbionts after exposure to the various treatments (i.e., toxicity, chilling, bleaching). All photosynthetic yield measurements were taken in the light after coral microfragments had sat in ambient lab lighting (approximately 2-4 PAR)

For the toxicity and chilling experiments, three different points on each microfragment were sampled at each of the 4–5 health assessment time points over time. For the bleaching experiments we used an Imaging-PAM, which is able to assess the photosynthetic yield across the entire microfragment. We chose to define five areas of interest (~1 mm2 each) with the Imaging-PAM, which allowed us to determine the photosynthetic yield of a much larger area of the microfragments. Based on preliminary and unpublished data, anything above a photosynthetic yield of 0.1, on a scale from 0 to 1, suggested functional symbionts.

Chilling sensitivity of coral microfragments and their algal symbionts

During a vitrification experiment, the coral might experience a certain period of chilling. Cells or tissues that are extremely sensitive to chilling can experience ruptures in the cell membranes around 0 °C. Therefore, we needed to explore the limit of what coral microfragments could tolerate and still recover from these exposures.

Preliminary experiments determined that coral microfragments could only withstand 1 min of chilling at 0 °C with no tissue loss or death (see Supplemental Methods and Data, Fig. S1). Lower temperatures (−10 °C) or longer exposures (2, 4, and 5 min) either led to immediate death, death within 2 weeks, or significant tissue loss. Therefore, microfragments (n = 11) were chilled at 0 °C for 1 min in cryovials with 1.0 mL FSW (0.22 µm filtered seawater) that had been pre-equilibrated to 0 °C for 10 min to determine how they would recover from this stress. After 1 min of chilling, microfragments were placed in 1 L of FSW at room temperature (~22 °C) for 10 min and then were placed in individual 6-well plates with approximately 10 mL FSW. For the first 24 h of culture, microfragments were kept in an incubator at 26 °C and 0 PAR and then given additional light (35–50 PAR) for 2 weeks.

Chilled microfragments were assessed at four time points: 24, 72 h, 1-, and 2-weeks post-chilling exposure. Each assessment included, 1) symbiont viability with a Junior PAM.; and 2) the integrity of the coral tissue with light microscopic imaging and health metric scoring (see Table 1 for details). After the 24 h assessment, microfragments were placed back into an incubator at 50 PAR and 26 °C. Coral microfragments were cultured in 6-well plates through the 72 h assessment where plates and water were changed daily. Afterwards, they were moved to 5 L aquaria with flow through FSW through the 2 week assessment at 35 PAR and 26 °C.

Growth response after chilling and cryoprotectants exposure

After the final health assessment, coral microfragments were secured to plastic sheets supported by plexiglass plates and suspended in the water column of an outdoor mesocosm in a flow through seawater system. Coral fragments were examined weekly to determine whether they had resumed normal growth and calcification. We defined growth by the production of one to two rows of coral polyps that form a full ring around the microfragment on the plastic sheet.

Microfragment bleaching

All reef-building corals have symbiotic algae, Symbiodiniaceae. The endosymbionts in P. compressa are from the genus Cladocopium (previously Symbiodinium Clade C), subclass C15 (Forsman et al., 2020; Krueger & Gates, 2012). Hagedorn et al. (2009) tested the cryophysiology of extracted symbionts from three different coral species in Hawai‘i, including Porites compressa, and determined that they all had similarly long permeation rates. Additionally, preliminary vitrification experiments by the authors confirmed that the permeabilities of the algal symbionts and coral tissue to cryoprotectants are very different (~1 h vs. 3 min, respectively). Prior to cryopreservation, trying to dehydrate and penetrate the algal symbionts with water and cryoprotectants, respectively, resulted in the coral fragment dying. Therefore, we developed bleaching protocols that maintained the coral health with the caveat that we would be able to reintroduce the algal symbionts after thawing during the culture period. Three treatments were preliminarily assessed to find the most rapid and least detrimental bleaching method-menthol (0.58 mM menthol; Wang et al., 2012), light (350 PAR for 17–18 h), and menthol and light (0.58 mM menthol + 350 PAR for 17–18 h).

Before the bleaching treatment, microfragments were imaged, assigned a health score (Table 2), and given a color rank assessed by Ko‘a Card (Bahr et al., 2020) in order to determine any change in health during the bleaching process. During the day, microfragments were placed in aerated, 2 L aquaria with 0.58 mM menthol (99%, Sigma Aldrich, St. Louis, MO, USA) in ethanol (190 proof; Decon Labs, Inc., King of Prussia, PA, USA) in filtered seawater with aeration for approximately 6–7 h, 26 °C, ~5 PAR. Then, they were transferred to a 26 °C incubator with 350 PAR for 17–18 h. The microfragments were cycled between the menthol bath and full light until they were fully bleached (i.e., had reached the lowest color ranking on the Ko‘a Card). This took approximately 72 h to 1 week. At the end of the bleaching treatment, microfragments were imaged on a light microscope with a Lumenera Infinity 3s camera, assigned a health score (criteria based on Table 2), given a color rank assessed by Ko‘a Card, imaged on a Zeiss LSM 710 laser-scanning confocal microscope, and the presence and viability of the algal symbionts was determined by assessing with a Walz Imaging-PAM.

Table 2 Health metric scoring criteria for bleached microfragments.

The health metric scoring for coral fragments that had been bleached were observed visually under a light microscope and given a score 0–5 using the criteria listed here.

Score	Description of metric for bleached microfragments	
0	Dead, tissue sloughing or >25% bacterial growth, >75% brown dots or sheeting of dead/dying algal symbionts on surface of coral	
1	>75% damaged polyps, <10% intact coenosarc, >50% brown dots or sheeting of dead/dying algal symbionts on surface of coral	
2	50% damaged polyps, 25% intact coenosarc, >25% brown dots or sheeting of dead/dying algal symbionts on surface of coral	
3	>75% intact polyps, 50% intact coenosarc, <10% brown dots or sheeting of dead/dying algal symbionts on surface of coral, no bacterial growth	
4	Intact polyps, 75% intact coenosarc, <5% brown dots or sheeting of dead/dying algal symbionts on surface of coral	
5	Intact polyps, 100% intact coenosarc, no brown dots or sheeting of dead/dying algal symbionts on surface of coral	

Confocal imaging of microfragments

Confocal imaging was used to quantify the success of microfragment bleaching by measuring mean fluorescent intensity of algal symbionts. Each coral microfragment was imaged using the Zeiss LSM 710 with a Zeiss Plan-Apochromat 5×/0.16 M27 objective. All microfragments were imaged with the same acquisition settings: z-stack: 12 slices, range = 330 um; image resolution: 2,048 × 2,048, (1,700 μm × 1,700 μm), 12-bit; pixel dwell: 1.57 microsec; pinhole size: 36 μm, 1 AU. The image frame for all samples was entirely composed of coral coenosarc tissue and one polyp. No blank space occupied the image frame.

The excitation wavelength of 405 nm was applied using a Diode laser at 15% intensity. Two channels were created to capture the autofluorescence of the coral microfragment. Channel 1 (515–575 nm) captured the autofluorescence of the coral host, Channel 2 (611–709 nm) captured the autofluorescence of the chlorophyll within the algal symbiont cells. The symbiont autofluorescence in the confocal image is used as a proxy for algal symbiont density within the coral tissue (Huffmyer et al., 2021).

For MFI analyses, Zen Black processing software (Zen 2.3 SP1 FP3 v.14.0.26.201) was used, all z-stack images were standardly formatted into maximum intensity projections. Fluorescence intensity data of the symbiont from the maximum intensity projections was pooled together within each experimental treatment (control and bleached) and averaged.

Statistical analyses

Measurements were represented by the means in all figures. All data were tested for normality and outliers using the ROUT method set at a sensitivity of 1.0%. For normally distributed paired data, parametric t-test were performed. If the data were not normally distributed, non-parametric tests (Mann-Whitney or Krukal-Wallis test) were done to test the differences amongst means. Where needed, analyses of variance (ANOVA) or Kruskal-Wallis tests were used to determine differences between groups (α = 0.05). When groups were significantly different, posthoc tests were conducted using Dunn’s multiple comparisons tests. All error bars in the figures are represented by standard error of the mean (SEM). Statistical analyses were conducted in Prism 9.31 (GraphPad, San Diego, CA, USA).

Results

Chilling and toxicity sensitivity of coral microfragments and their algal symbiont

We examined the chilling sensitivity of the P. compressa microfragments at 0 °C for 1 min exposures over 2 weeks (Fig. 1). Untreated (positive control) microfragments maintained a uniform health metric of five throughout the treatment period (Kruskal-Wallis; p > 0.05), and their mean photosynthetic yield did not vary greatly, although the initial readings at 0 and 24 h were lower than the later values over the 2 week period in culture (Kruskal-Wallis H(6) = 17.05; p < 0.001; Dunn’s Multiple Comparison test). Compared to the control values at 0 h, the health metrics of the chilled microfragments (Figs. 1A and 1B, blue bars) showed a decline followed by an improvement at 2 weeks, with the 2 week recovered microfragments comparable to control values (Kruskal-Wallis H(5) = 17.33; p < 0.001; Dunn’s Multiple Comparison test). During this recovery period, chilling caused a loss of coral and algal symbiont cells, which were observed surrounding the microfragments in the culture dishes for up to 72 h. These stressors caused the microfragments to display a pale coloration before they recovered. At 72 h, the photosynthetic yield was lower than the control values, after which it returned to pre-treatment values (Kruskal-Wallis H(5) = 17.09; p < 0.004; Dunn’s Multiple Comparison test).

Figure 1 Average health metric score and photosynthetic yield for microfragments exposed to cryogenic stressors.

(A) Bar graph of health metric score for microfragments exposed to chilling, toxicity, nothing (control), and a cryo-damaged or dead microfragment. (B) Bar graph of the photosynthetic yield for microfragments exposed to chilling, toxicity, nothing (control), and a cryo-damaged or dead microfragment. * Statistically significant when compared to the control.

Microfragments exposed to vitrification solution demonstrated some loss of coral cells and algal symbionts which was associated with some tissue retraction at the 24 h to 2 week time-period reflecting poorer health status, followed by a recovery at 3 weeks (Figs. 1A and 1B, gray bars; Kruskal-Wallis H(6) = 57.08; p < 0.0001; Dunn’s Multiple Comparison test). There was no visible loss of algal symbionts observed during any cryoprotectant treatment, however, the photosynthetic yield was slightly lower at the 24, 72 h and 3 week time points (but not at the 1 week and 2 week time points) when compared to the 1 week control photosynthetic yields (Kruskal-Wallis H(5) = 12.21; p = 0.0159; Dunn’s Multiple Comparison test). These experiments were critical to understand the physiological changes that the coral might undergo prior to cryopreservation, during which they will be exposed to low temperature stressors, as well as toxicity.

Growth response after chilling and cryoprotectant exposure

After the coral microfragments were exposed to chilling or toxicity, they were kept in recovery for 2 to 3 weeks in the lab and then returned to running seawater tanks to determine how long it would take them to begin growing and calcifying. Not all microfragments were followed through re-growth because they were dislodged from the plastic sheeting. The sample size for each treatment was as follows: (1) no treatment (n = 9), (2) chilled to 0 °C for 1 min (n = 9), or (3) exposed to VS for 6 min (n = 7). The mean time for each group to begin growing in the seawater system was 2 months and there was no difference between any of the treatments (Kruskal-Wallis test; p > 0.05), suggesting that once the microfragments had recovered in the laboratory for 2 to 3 weeks, their previous treatment did not affect their future growth. Specifically, the timing for re-growth was Control = 60.0 ± 6.3; Chilling = 65 ± 5.1; Toxicity = 56.1 ± 5.0 days.

Confocal imaging

Confocal imaging was assessed as a tool to (1) differentiate between live and dead tissue by looking at the fluorescence of the coral fragment, and (2) quantify the success of microfragment bleaching. Living coral microfragments have a well-defined distribution of the auto-fluorescent green fluorescent protein (GFP), and a discrete distribution of their auto-fluorescent algal symbionts, which generally surround the tentacles and polyp mouth. However, each genotype has a unique distribution of these fluorescent signatures that define these living microfragments. The control pattern of fluorescence for live and dead coral is shown in Fig. 2, where the dead coral has a different fluorescent pattern. After a coral fragment has gone through several freeze-thaw cycles, ice crystals disrupt their membranes, and the microfragments die. This causes the GFP and algal symbiont fluorescent signals to become disaggregated and disorganized, producing a smeared appearance, although the GFP signal remains up to 72 h (Fig. 2). In fact, when the mean fluorescence intensity of the live and dead corals was compared, there was no difference in GFP or algal symbiont fluorescence. Because of the longevity of the GFP in the tissue, the presence of this signal was deemed to not serve as an appropriate indicator, and therefore, could not be used to quantify viability of post-thaw coral microfragments.

Figure 2 Confocal images of a live and dead coral fragment.

Confocal images of a polyp from a live and dead coral fragment after 24 h in culture. (A) Live polyp, 24 h: Merged image of the autofluorescent symbiotic algae (red) and the autofluorescent green fluorescent protein of the coral (GFP, green). Note how tightly organized the GFP and symbionts are around the polyp mouth and tentacles. The confocal image clearly shows the morphology of the polyp skeleton and tentacles. (B) Dead polyp, 24 h: Merged image of the autofluorescent symbiotic algae (red) and the autofluorescent GFP of the coral (green). Note the disorganized pattern of the GFP and symbionts fluorescence around the polyp and tentacles. The polyp skeleton and tentacles were degraded and appear blurred. The symbiont fluorescence is scattered across the image and the GFP is blurred.

Confocal imaging was used to determine the success of intentional microfragment bleaching to assess whether the symbionts disappear from the tissue or were non-functional. Preliminary experiments determined that a combined menthol and light bleaching treatment resulted in an 83% decrease in mean fluorescence intensity between the wavelengths fluoresced by the algal symbiont after 72 h of exposure (Fig. 3). Specifically, in this image, the control had a mean fluorescence intensity = 284.6; light treated = 78.2; menthol = 89.1, and menthol and light = 42.1, and as shown in Fig. 3 (n = 1, preliminary data). Additionally, all treatments maintained a health metric score of 5, therefore, the menthol and light bleaching treatment was used for subsequent assessments.

Figure 3 Light and confocal images that illustrate the effects of various ‘bleaching’ treatments on coral microfragments.

Coral microfragments were treated for 3 days with one of three bleaching treatments: light, menthol, and menthol & light. Additionally, the microfragments were imaged on a confocal microscope to assess the density of symbionts. (A) Light images of coral microfragments that were ‘bleached’ using menthol, light, and menthol & light. (B) Confocal images of the same coral microfragments but at the polyp level. The merged images layer the GFP (green) and autofluorescence of the algal symbionts (red) into one image. Preliminary data show that the combination of Light & Menthol reduced the density of symbionts the most.

After preliminary data showed that menthol and light bleaching was effective, menthol and light bleached microfragments (n = 25), were imaged with confocal microscopy, Imaging-PAM, and given a health metric score to determine whether this treatment would significantly reduce the algal symbiont population without seriously compromising the health of the coral. Coral microfragments were bleached for 72 h to 1 week and the loss of their symbionts was monitored. Microfragments from the same genotypes (n = 25) were left untreated or bleached with menthol and light (Fig. 4). These treatments demonstrated a 78% loss in the mean fluorescence intensity (MFI) or the number of symbiont-like fluorescing particles (Controls = 212.1 ± 19.6; Bleached = 47.0 ± 2.5; two-tailed t-test, t(24) = 8.90, p < 0.0001).

Figure 4 The loss of the symbiotic algae was monitored with confocal microcopy (MFI) and Imaging-PAM (photosynthetic yield).

Coral microfragments were bleached up to 1 week and the loss of their symbionts was monitored with confocal microscopy and Imaging-PAM. (A) Paired microfragments from the same genotypes (n = 25) were left untreated or bleached with menthol & light. These pairs demonstrated a 78% loss in in the Mean fluorescence Intensity or the number of symbiont-like fluorescing particles (Controls = 212.1 ± 19.6; Bleached = 47.0 ± 2.5). (B) When bleached and untreated microfragments from the same genotypes (n = 10) were examined with an Imaging-PAM, a 98% loss in photosynthetic yield (Y) was observed. The control fragments had a mean Y-value of 0.567 ± 0.006, whereas the bleached values were reduced to 0.013 ± 0.007, suggesting that none of the remaining symbiont-like particles in the bleached fragments were functional. Means with an asterisk (*) were different p < 0.001, paired parametric t-test (A) and nonparametric non-paired Mann-Whitney U test (B), all errors represented by SEM.

Bleached and untreated microfragments from the same genotypes (n = 10) were examined with an Imaging-PAM, and a 98% loss in photosynthetic yield (Y) was observed. The control microfragments had a mean photosynthetic yield value of 0.567 ± 0.006, whereas the bleached values were reduced to 0.013 ± 0.007 (Wilcoxon matched-pairs test, p = 0.0039), suggesting that none of the remaining symbiont-like particles observed in the bleached microfragments with confocal microscopy were functional.

Each microfragment was assessed under a dissecting microscope (Olympus SZXILLB2100, magnification 20×) and given a score based on the rubric in Table 1 (control/unbleached) or Table 2 (bleached). Thus, the combined use of menthol and light reduced symbiont distribution and function throughout the coral tissue and maintained good health metric scores, 4.0 ± 0.3.

Discussion

The overarching aim of this study was to understand and assess the physiological stressors that might underlie coral microfragment cryopreservation and the long-term consequences of these physiological exposures to continued coral growth in a land-based nursery. Coral microfragments initially responded negatively to chilling and toxic conditions but recovered over a 2 to 3 week period in the laboratory. After the microfragments had recovered, they were placed in outdoor tanks; they began to grow within 2 months, and their previous treatment did not affect their future growth. This may indicate that cryopreservation will cause some short-term stress, and will not cause any long-term physiological impacts for the coral. To aid these findings, we developed a scale of visual health metrics (a “Health Metric Score”) for these experiments based on preliminary data of severely damaged corals (e.g., chilling and toxicity) because other health assessment tools did not accurately assess damage and recovery. Moreover, we did not find anything in existing literature that could accurately quantify damaged coral based upon visual qualitative observations of varying levels of chilling and toxicity exposure.

The results from chilling for 1 min at 0 °C showed that it took microfragments 2 weeks to recover fully (i.e., statistically the same health metric score as the live controls, Fig. 1). Visually, they lost many of their symbionts within 24 h but this was not reflected in the PAM data (Fig. 1). It took the microfragments exposed to the vitrification solution, 3 weeks to recover completely (Fig. 1). Like chilling, they lost many of their symbionts within 24 h. Initially, during preliminary experiments, microfragments exposed to CPAs in toxicity treatments appeared to be dead and it was unexpected when they began to recover after 2 weeks. These preliminary observations were the reason corals exposed to CPAs were given 3 weeks to recover rather than 2 weeks, like the corals exposed to chilling.

Menthol and light bleaching resulted in healthy corals (health metric score 4.0 ± 0.3) that were visibly bleached (Fig. 3) and functionally bleached (Fig. 4). This finding was significant because it is likely that the coral and its symbiotic algae cannot be cryopreserved together because of difference in CPA loading requirements (Hagedorn et al., 2009). Due to their greatly different permeation rates, they cannot survive the whole cryopreservation process if treated as one. Therefore, future work will likely include bleached microfragments in order to test the hypothesis that the coral alone may survive the cryopreservation process better than combined with the symbiotic algae.

It was expected that the GFP intensity would decrease in intensity more quickly over time in the dead microfragments such that GFP might be used as a post-thaw indicator of viability. However, the confocal imaging demonstrated that this was not the case; instead of decreasing in intensity, it remained constant and, in some cases, showed a higher level of intensity than the live controls. Additionally, the pattern of GFP in and around the coral polyp changed from a very distinct rosette pattern to a more diffuse, distributed pattern. In contrast, another study found that warmed and cooled coral (±5 °C) did demonstrate a loss of symbionts and GFP signal intensity (Roth & Deheyn, 2013), so the persistence of the GFP signal in the dead coral used in this study was both surprising and, ultimately, not did not serve as a useful indicator of post-thaw viability for future coral cryobiology studies.

Even healthy corals that have not gone through cryopreservation can be difficult to assess and compare their “healthy status”. Health metrics common to vertebrate organ systems are virtually nonexistent for invertebrates, and as sessile animals, coral movement is limited to polyp extension and contraction, with skeletal malformations, tissue integrity, and tissue coloration serving as other visual indicators. The process of cryopreservation causes stress due to chilling and toxicity damage, and it was not possible to refine our cryopreservation process without a standardized health assessment key derived from cryobiology-related damaged coral tissue. Before we created the health metric scoring assessment in this study, we used confocal imaging and PAM fluorometry to attempt to determine the degree of damage or stress. We used confocal microscopy to assess the GFP native to coral tissue as a proxy for coral health, assuming that the GFP would gradually decrease in damaged or dead coral. Unfortunately, GFP is very persistent and was approximately equal and sometimes greater in the dead corals even after 72 h (preliminary data). Additionally, we tried to use PAM fluorometry–which measures the photosynthetic potential in the symbiotic algae and is often used by coral biologists as another proxy for coral health–for assessments. PAM fluorometry was also not a good indicator because the signal (photosynthetic yield) did not incorporate the loss of symbiotic algae, and in severely damaged or dead microfragments, other photosynthetic organisms and algae colonized the coral skeleton inaccurately skewing the results. Therefore, we developed the health metric scores in Table 1 (for unbleached microfragments) and Table 2 (for bleached microfragments). The health metric systems developed for this study allowed us to visually assess and compare treatments along a spectrum of coral tissue damage common and applicable to cryobiology stress where other methods proved unreliable.

A promising avenue to successfully cryopreserve an organism as large as a coral microfragment is a technique called isochoric vitrification. To date, most biological matter is cooled under constant pressure or isobaric conditions at atmospheric pressure, which allows the sample to change volume. Moreover, only samples of relatively small size (~100 µm in diameter), such as a human embryo, can be easily vitrified using these methods. However, emerging techniques aim to preserve biological material at constant volume (Rubinsky, Perez & Carlson, 2005), confining the system and denying it access to the atmospheric pressure reservoir. This isochoric cryopreservation process generally employs only a single step–cooling whereby the system is in a perfectly confined, constant-volume chamber. The technology does not require moving parts, mechanical work, and can be used on much larger samples. Therefore, isochoric vitrification may be ideal for processes involved in field cryopreservation of coral microfragments near or on reefs.

Given the demonstrated sensitivity of coral microfragments to mild chilling temperatures, this study suggests that isochoric vitrification may present a suitable technique to successfully cryopreserve them. How might this new vitrification process work for coral microfragments? According to this study, corals could withstand chilling temperatures up to 1 min, and complex vitrification solutions for up to 6 min. Using isochoric vitrification, small volumes of vitrification medium (~5 mL) can be frozen to liquid nitrogen temperatures (−196 °C) in less than 2 min. The amount of time where the microfragments might remain at chilling temperatures (0 to −10 °C) either on cooling or warming is less than 20 s, well below the 1 min threshold found for chilling. Furthermore, previous work (Zhang et al., 2018) has found that isochoric confinement can reduce the CPA concentrations required to vitrify a given solution (as compared to conventional isobaric vitrification), which may enable use of minimally toxic cryoprotective solutions that would otherwise be susceptible to destructive ice formation.

Interventions are needed to help secure and restore coral reefs (National Academies of Sciences, Engineering and Medicine, 2019) and coral cryopreservation of all types is a developing tool to aid in these conservation actions. However, engineers, coral biologists and cryobiologists must partner to help develop the tools to cryopreserve larger and more complex cells and tissues to usher the field into the future. In terms of complexity of the tissues and the differences in the cryo-permeabilities of coral holobiont, the cryopreservation of coral microfragments might be considered almost as complex as human organs, such as an embryonic kidney or heart. Toward this end, there is emerging technology that may be a good candidate for coral. Restoration processes might benefit from the development of coral microfragment cryopreservation by allowing the safe preservation and reanimation of hundreds of thousands of small microfragments potentially encompassing many of the coral species in the wild. Moreover, the amount of space required for these cryopreserved coral assets is minimal, especially when compared to bringing live coral assets into captivity. For example, when stony coral tissue loss disease hit the Florida reef tract, many vulnerable corals were brought into captivity and it took a consortium of zoos, aquariums, and universities to be able to house the 2,500 colonies (Florida Sea Grant, 2022).

Some recent models suggest that time is limited for these types of intervention processes (Dixon et al., 2022; Kalmus et al., 2022). If the cryopreservation of coral microfragments is to be successful, there remain many unanswered questions about how many microfragments must be preserved and when and where they might be collected. However, there is still time for the scientific community to develop the technology fully, train professionals and bank the biodiversity in our oceans. Novel ex situ conservation strategies, such as genetic biorepositories holding cryopreserved coral microfragments, hold strong promise to help offset many of the anthropogenic threats facing coral reefs today.

Supplemental Information

Supplemental Information 1 Supplemental methods and data.

Supplemental Information 2 Health metrics of coral fragments.

Raw dataset

The authors would like to thank Dr. Shayle Matsuda for his advice and guidance on bleaching coral. We would like to thank the Coral Resilience Laboratory for the generous use of their Imaging-PAM. We would like to thank our summer interns: Kendall Fitzgerald and Morgan Brooks for their assistance in microfragment husbandry. HIMB contribution #1972 and SOEST contribution #11855.

Additional Information and Declarations

Competing Interests

Author Contributions

Data Availability

Matthew J. Powell-Palm Is an Associate Editor for PeerJ.

Claire V. A. Lager conceived and designed the experiments, performed the experiments, analyzed the data, prepared figures and/or tables, authored or reviewed drafts of the article, and approved the final draft.

Riley Perry conceived and designed the experiments, performed the experiments, authored or reviewed drafts of the article, and approved the final draft.

Jonathan Daly conceived and designed the experiments, authored or reviewed drafts of the article, and approved the final draft.

Christopher Page performed the experiments, authored or reviewed drafts of the article, and approved the final draft.

Mindy Mizobe conceived and designed the experiments, authored or reviewed drafts of the article, and approved the final draft.

Jessica Bouwmeester analyzed the data, authored or reviewed drafts of the article, and approved the final draft.

Anthony N. Consiglio analyzed the data, authored or reviewed drafts of the article, and approved the final draft.

Jake Carter conceived and designed the experiments, authored or reviewed drafts of the article, and approved the final draft.

Matthew J. Powell-Palm conceived and designed the experiments, authored or reviewed drafts of the article, and approved the final draft.

Mary Hagedorn conceived and designed the experiments, analyzed the data, prepared figures and/or tables, authored or reviewed drafts of the article, and approved the final draft.

The following information was supplied regarding data availability:

The raw data are available in the Supplemental Files.

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
