# Peer review of "Cryophysiology of coral microfragments: effects of chilling and cryoprotectant toxicity"

_PeerJ, doi:10.7717/peerj.18447_

## Round 0.1 · original submission · Major Revisions

Dear authors,

Your manuscript submission has been peer reviewed by two experts in the field and both reviewers are of the same opinion that your submission requires significant improvement before it can be considered further in the PeerJ system.

Post reviewing the reviewers' extensive comments, and the manuscript itself, I can only but agree with the decision of the two expert reviewers. Once you have prepared a significantly improved version of your manuscript, I also required a point-by-point letter of response to the each of the reviewers' concerns / comments to be submitted along with a revised manuscript.

Considering the degree of concerns / comments raised by the two reviewers, please do take considerable time addressing all concerns / comments made by the reviewers before submission of your revised manuscript.

I am looking forward to receiving your substantially improved manuscript version in the future.

Kind regards,
Andrew Eamens

Reviewer 1 ·

Basic reporting

Lager et al. have tested the sensitivity of Porites compressa microfragments to chilling and cryoprotectant toxicity, as well as the feasibility of actively bleaching to separate host and algal components of the holobiont. Overall, they present important considerations for the preliminary assessment of cryopreservation preparation in corals beyond gamete-based cryopreservation. The manuscript is generally well-written and asks a unique question and provides a different perspective than what one normally considers when thinking about cryopreservation. The manuscript could be strengthened with 1) additional incorporation of references to support certain assertions, 2) clarification/more detail in the methods, 3) a clear statement regarding the limitations of this microfragment approach, and 4) a reframing of the discussion to focus more specifically on the study findings. Having made these changes, I believe the manuscript would be suitable for publication in PeerJ.

Experimental design

Samples size (n = 11) is quite low; because of this the authors should clearly acknowledge the limitations of extrapolating findings based on a small sample size.

Validity of the findings

No comment.

Additional comments

Title: Please change to a title that is more descriptive of your study

Abstract:

The abstract does not include any mention of results. Please reduce the text describing the methods and include a concise summary of your results. This will allow readers to quickly understand the key findings/context of your manuscript.

Line 27: “to help them” sounds quite informal, perhaps change to ‘improve coral health and persistence’ (or something like that).

Line 31: Change “these physiological exposures” to ‘exposures’ or ‘low temperature exposures’.

Line 32: should this be mm, not mm2?

Line 33: change “fragments” to ‘coral’


Introduction:

The first 2 paragraphs could be deleted (the sentences jump around a bit and don’t necessarily lead the reader clearly into your topic of interest); starting the introduction on line 56 presents a more streamlined approach.

Line 64: try to avoid terms like ‘so perilous’, and stay keyed in on a coral focus, i.e., do not broaden out again by stating ‘loss of species in our oceans’.

Line 69: Please add in some more recent references here.

Line 78: ‘new coral’ seems like you are implying a new species; please rephrase.

Line 82: Likely should mention that you are referring to one specific reproductive strategy in corals (broadcast spawning).

Line 84: Please add references that specifically back up the first part of the sentence.

Line 87: It is not clear how microfragment preservation is “immune to climate change issues” (source colonies will still likely be impacted by climate change); please rephrase.

Line 92: Could you add in a mention/reference here of another model organism that has a clear husbandry pathway from freezing to post-thaw survival/growth?

Line 93: Change “paper” to ‘study’

General comment: You do not mention egg cryopreservation anywhere; the reason for this is because it hasn’t (to the best of my knowledge) been accomplished, but the challenges of egg cryopreservation should likely be mentioned somewhere very briefly in the introduction. This will simply add more context as to why the proposed microfragment approach merits further investigation.


Methods:

Line 109: Please write our Hawaii in full, not abbreviated as HI for international readers. Also, likely important to mention in the methods that this is an endemic species (correct?)

Line 112: How many colonies? Please specify what is meant by “care”, 5 m apart, or further?

Line 116: How long did colonies recover for before fragmentation? How were the fragments cut (what type of saw was used)?

Line 121: Add ‘vitrification’ to ‘solution preparation’ heading

Line 129: Please explain why solutions were reduced compared to other studies; the rationale for this choice is not clearly stated.

Line 141: Were measurements taken in the light or were they dark adapted? Are you measuring Fv/Fm? If yes, please state this in the text and your figures.

Line 151: add ‘s’ to symbiont

Line 161: A sample size of 11 is quite small; were there replicates included for each of these 11 genotypes?

Line 165: Why was PAR 0? Can you point to other studies that use a similar approach? It seems that this adds in another variable (i.e., lack of light) that may confound your overall findings. This comment also applies to Line 182.

Line 171 and 173: How do these PAR levels match up to what the colonies would normally experience on the reef?

Line 176: Were there 11 fragments in each solution treatment? As it is written it sounds like the 11 fragments were spilt among the treatments.

Line 187: Were these health scores defined by the authors or based on other studies, if based on other studies please provide a reference.

Line 208: Is the genus or species of algal symbiont(s) known for Porites compressa based on other studies? If yes, please mention this somewhere in this section.

Line 217: Please indicate the samples size for the bleaching experiments.

Line 233: Is ‘post thaw’ an accurate description? The temperature tested does not go below zero in the chilling experiment. That is what the authors are referring to here, right?

Line 261: Please write SEM in full.


Results:

In general, I have some concerns about the light levels used in these experiments and how they may have affected the overall findings. This doesn’t mean that the author’s work is not valid/useful, but there needs to be some acknowledgment that the responses in algal symbiont performance that are being observed may also be a response to low light exposure in the experiments (likely lower than typically experienced on the reef, correct?). Additionally, some of the text in the results seems like it would be more appropriately placed in the methods; please rephrase to focus on just reporting the results in this section and place explanations/background information in the methods (e.g., lines 248-350 would be more appropriate in the methods section – I don’t think there was any mention of a paired design in the methods, this is first mentioned in the results?).

Line 267: Change “internal” to algal to match terminology elsewhere in the manuscript.

Line 275: Change “true measure” to a ‘more accurate measurement’ (or something like that)

Line 310: Why were not all fragments used? Mortality? If yes, this needs to be clearly stated.

Line 336: Delete “Additionally”

Line 341: Please include standard deviation or error with the mean values to give the reader an indication of variability in your results.

Discussion: Overall, the discussion is very well written and introduces the concept and potential applicability of isochoric vitrification effectively. This manuscript, however, is not specifically testing isochoric vitrification and the discussion does not robustly ‘discuss’ the manuscript’s key findings. I suggest reframing the discussion to relate it more toward the interesting study findings and limitations, and then move on to introducing the applicability of isochoric vitrification.

Line 386: Change “cryopreserved” to ‘cryopreserve’


Figures & Tables:

Figure 1: I am confused by the ‘dead’ bars. Please explain. Which treatment are these data referring to?

Figure 2 & 3: Please add scale bars.

Table 2: What you mean by “freckles” is unclear, please rephrase or define.


Overall, this was an interesting manuscript to read and was no doubt a product of a lot of hard work by the authors. I wish you all the best with your revision and the future publication of this manuscript.

Reviewer 2 ·

Basic reporting

In their study, Lager et al explore the impact of cold exposure and cyroprotectants on microfragmented corals. Cold exposure and the toxicity of cryoprotectants may be detrimental to coral health, but my and the authors argue that understanding these impacts will be important for developing effective cryopreservation techniques to store coral long-term. This has been accomplished with gametes, but my understanding is that cryopreservation of coral larvae, much less larger chunks of somatic tissues of corals is still in its infancy. The authors argue that being able to preserve coral stock from gametes to fragments of adult colonies that can be grown again after thawing will be important to safeguard the genetic diversity of coral that are currently under threat from global climate change. I agree with the authors on this premise and think this is important work, especially considering the small spatial footprint cryocollections would have compared to the alternative of living ex situ collections in aquaria.

The authors investigated the effects of cold shock and cryoprotectant toxicity separately, which confused me at first. In the discussion, the authors make the point that understanding the effects of either cold shock or toxicity need to be understood prior to developing a full protocol for coral cryopreservation via vitrification. I feel like I had to read to the end of the paper to get this point, which should be made much earlier in the introduction. While the last paragraph of the introduction mentions the needs of understanding the stressors that come with cryopreservation protocols, the authors should be clearer about their aims in the introduction. My suggestion is to condense the lengthy discussion on climate change on coral reefs to a few sentences, citing the appropriate sources. Some of the background information included in the discussion should be moved into the introduction. That is, a short review of cryopreservation methods and the state of the field for coral preservation. Vitrification should be described in the introduction and the general procedural flow of this technique. Are there other systems in which this technique is successful? What are the types of tissues or organs that can be preserved in this way? How does cold shock and toxicity impact tissues during sample preparation for vitrification? I assume that during the preservation process, both are experienced for some amount of time; the current paper investigates what those limits are.

The language on coral reproduction in the introduction seemed disconnected. The clearest connection to the current work seems to be that cryopreservation protocols exist for coral germ cells. However, coral reproduction has been suppressed by rising temperatures and for many corals, spawning times may not be known. In general, collecting germ cells for preservation requires access to the reef at a specific time. So the proposal to cryopreserve small fragments of coral that could be collected at any time seems sound to me.

Experimental design

Overall, the experimental design seems sounds and the data collected appear appropriate for the questions addressed. My concern with the paper in its current form is more one of presentation. The experimental design is somewhat hidden and scattered throughout the manuscript. I indicated in my general reporting that, in my opinion, the paper would benefit from a bit of restructuring, especially providing a better overview of the field in the introduction and identifying gaps that the current paper addresses. The authors should then explicitly describe their experimental design. In the current paper, the authors state what they did at the end of the introduction but a clearer experimental design layout would help better understand how these data address the broader question.

Line 193-198 contain some key information on experimental design that should be rephrased somewhat and mentioned far earlier in the paper to help readers understand the concept of the tests that were conducted. In particular, do I understand correctly? Your point is that toxicity is an issue at room temperature because there is an incubation step in cryprotectant solution (don’t use the term vitrification medium; it caught me off-guard but I think you mean cryoprotectant) at room temperature. So toxicity at RT is of concern. Cold shock plus cryprotectant were not used as a treatment since chilling of the sample would occur after it has been exposed to cryprotectant already. That seems reasonable. I would suggest moving some of this text into the introduction or into an experimental design section early in the materials and methods. The reader should have a broad overview of the experiment and why it was designed in this particular way early on.

Validity of the findings

Overall, the methodology appears sound for the questions addressed. The authors used a combination of microscopy, scoring of morphological traits similar to what is being done in field surveys, and fluorometry to investigate in the impacts of cold shock and toxicity on coral fragments. The data of post treatment growth and symbiont photosynthetic performance suggests that the treatment did not affect the coral fragments long term. So this seems to be a promising step in the direction of developing a full cryopreservation protocol for coral fragments.

Additional comments

l. 117: If a specific type of superglue was used, provide brand name and manufacturer. Alternatively, consider using the more general term cyanoacrylate adhesive.

l. 121: Be specific which solution you are referring to. You mean a cryoprotectant, correct?

ll. 124-126: You describe the general process. Is there a review paper that could be cited?

l. 141: photosynthetic yield: be more specific. I assume this is maximum quantum yield (Fv/Fm) of dark adapted fragments. Describe what this measures and cite an appropriate source. I would also suggest to change all axis labels on figures to max quantum yield or Fv/Fm if this is what is measured.

l. 150: Why was the cutoff 0.1? I assume you are trying to distinguish noise from signal. Is there a source that you can refer to for the cutoff? Your own data of dead fragments suggests that 0.1 is on average a reading that could be expected from non-living material. So the cutoff seems sound but should be justified in the methods.

l. 176: VS75 and VS80 are referred to as VS 75% and VS 80% earlier. I suggest using consistent terminology.

l. 212: I do not understand what you mean by trying equilibrate the algal symbionts. Is this appropriately described in the methods?

l. 216: What does light treatment mean? In the results, PAR measurements are provided but it should be made explicit here what the light levels for the treatments are.

Table 3: Column labels "winter" and "summer": did I miss this in the text elsewhere? I am not sure what these labels refer to.

---

## Round 0.2 · Minor Revisions

Dear authors,

I really like your study and find its reported data highly interesting and currently very topical.

However, quite a lot of textual issues remain in your manuscript and must be addressed before moving this study on.

Please address all of my comments/concerns via using the annotated review of your study's PDF as a template for required improvement.

All the best,
Andy Eamens

Reviewer 2 ·

Basic reporting

I read the responses of the authors to the reviews and their revised manuscript. The authors addressed the reviewer's comments and reorganized sections of their paper, as suggested. In my opinion, the paper is now much improved and clearer to follow. I also appreciate the change of the title which now more accurately reflects the content of the paper. In its revised form, the manuscript should make a useful contribution to the field.

Experimental design

The design seems appropriate for the study.

Validity of the findings

One weakness is the overall low sample size. Overall, the results appear compelling despite this issue.

Additional comments

A few minor comments below using line numbers from the marked up manuscript:

ll. 42-43: "statistically comparable": this should be rephrased to indicate that treatment samples were not significantly different from controls.

ll. 96-97: I am not sure about PeerJ's policy on this but the URL pointing to an online article should probably be added to the citations with a date when the article was accessed.

ll. 152-154: Company names and their locations (City, State/Country) should be provided for each instrument or supply item. There are a few other places throughout the manuscript where this information is missing and should be added.

---

## Round 0.3 · accepted · Accept

Dear authors,

Thank you kindly for your extensive revisions made to your study.

And, thank you for working so cooperatively in the previous rounds of requested review - this is very much appreciated also.

I believe your manuscript is now at the standard suitable for publication acceptance in PeerJ - congratulations.

Kind regards, Andy